# Reversed thermo-switchable molecular sieving membranes composed of two-dimensional metal-organic nanosheets for gas separation

Xuerui Wang[1], Chenglong Chi[1], Kang Zhang[1], Yuhong Qian[1], Krishna M. Gupta[1], Zixi Kang[1], Jianwen Jiang[1] & Dan Zhao[1]

It is highly desirable to reduce the membrane thickness in order to maximize the throughput and break the trade-off limitation for membrane-based gas separation. Two-dimensional membranes composed of atomic-thick graphene or graphene oxide nanosheets have gas transport pathways that are at least three orders of magnitude higher than the membrane thickness, leading to reduced gas permeation flux and impaired separation throughput. Here we present nm-thick molecular sieving membranes composed of porous two-dimensional metal-organic nanosheets. These membranes possess pore openings parallel to gas concentration gradient allowing high gas permeation flux and high selectivity, which are proven by both experiment and molecular dynamics simulation. Furthermore, the gas transport pathways of these membranes exhibit a reversed thermo-switchable feature, which is attributed to the molecular flexibility of the building metal-organic nanosheets.

[1] Department of Chemical & Biomolecular Engineering, National University of Singapore, 4 Engineering Drive 4, Singapore 117585, Singapore. Correspondence and requests for materials should be addressed to D.Z. (email: chezhao@nus.edu.sg).

Previous studies on ultrathin two-dimensional (2D) membranes are primarily focused on 2D inorganic nanosheets as building blocks, such as exfoliated zeolites[1–3], graphene[3–5], or graphene oxide (GO) (refs 5–9). These building blocks with strong mechanical strength can be easily obtained for membrane fabrication. However, the resultant membranes are greatly limited by the availability and tunability of the inorganic building blocks[10–13]. Recently, Yang and co-workers demonstrated the nm-thick molecular sieving membranes composed of exfoliated zeolitic imidazolate framework (ZIF) nanosheets exhibiting a tenfold increase in gas permeation flux compared to GO membranes[6,14]. Nevertheless, the ZIFs that can be easily exfoliated into 2D nanosheets are still rather limited[15]. Compared to ZIFs, metal-organic frameworks (MOFs) composed of carboxylate ligands offer a much wider choice in terms of structural diversity, pore geometry and functionality[16–19]. The MOF nanosheets hence can serve as promising building blocks for 2D molecular sieving membranes[20–23]. Nevertheless, no such molecular sieving membrane has been reported yet because of the elusive synthesis of intact, non-aggregated and ultra-large MOF nanosheets so far. This can be attributed to the structure deterioration or fragmentation of MOF nanosheets during exfoliation[24,25] and membrane fabrication[26], which can be seen by the fact that the elastic modulus of 2D MOFs (3–7 GPa) (refs 25–28) is much lower than that of other 2D inorganic nanosheets such as monolayered graphene (1,000 ± 100 GPa) (ref. 29) and GO (207.6 ± 23.4 GPa) (ref. 30).

In this study, we exfoliate a layered MOF, MAMS-1 (Mesh Adjustable Molecular Sieve, $Ni_8(5\text{-bbdc})_6(\mu\text{-OH})_4$) (ref. 31), into nanosheets and fabricate 2D membranes based on them. MAMS-1 is chosen due to its excellent hydrothermal stability and 2D layered structure (Fig. 1a,b). The atoms in each layer are connected through robust covalent and coordination bonds, while the layers are held loosely together through van der Waals interactions which can be easily overwhelmed to afford nanosheets. Each monolayer of MAMS-1 nanosheet, with a thickness of ca. 1.90 nm, possesses two possible gas permeation pathways interconnected with each other. The first pathway (PW1) lies roughly along the [001] direction (nearly perpendicular to monolayer basal plane) and has an aperture size of ca. 0.29 nm on both sides of the monolayer (Fig. 1c,d). The second pathway (PW2), with an aperture size of ca. 0.50 nm, is incorporated within the monolayer and distributed along the [100] direction (parallel to monolayer basal plane, Fig. 1e). Given the special configuration of gas pathways and the small aperture size of PW1, it is anticipated that the 2D membranes obtained by aligning the exfoliated MAMS-1 nanosheets along the [001] direction should be able to separate gases via molecule sieving mechanism. In addition, the aperture of PW1 gated by two pairs of *tert*-butyl group (Fig. 1d) may demonstrate certain stimuli-responsive properties due to the dynamic rotation of these groups[32–34]. On the other hand, the relatively large aperture size of PW2 can allow a quick redistribution of the permeated gas molecules (for example, $H_2$) throughout the whole monolayer, resulting in enhanced gas permeation flux. Furthermore, the hydrophilic inner wall of PW2 will impose different affinity towards various gas molecules (for example, $CO_2$ versus $H_2$) leading to improved separation performance[35,36].

## Results

**Preparation of exfoliated MAMS-1 nanosheets.** 2D MOF nanosheets can be readily obtained by top-down strategies (for example, sonication, ball milling, and so on)[14,24–26] but will easily result in large portions of fragmentation due to their low elastic modulus mentioned above. Consequently, the separation selectivity of fabricated membranes would be impaired because of the uneven alignment of 2D MOFs nanosheets in the membrane layer[37]. Herein, we adopt a mild exfoliation strategy to exfoliate

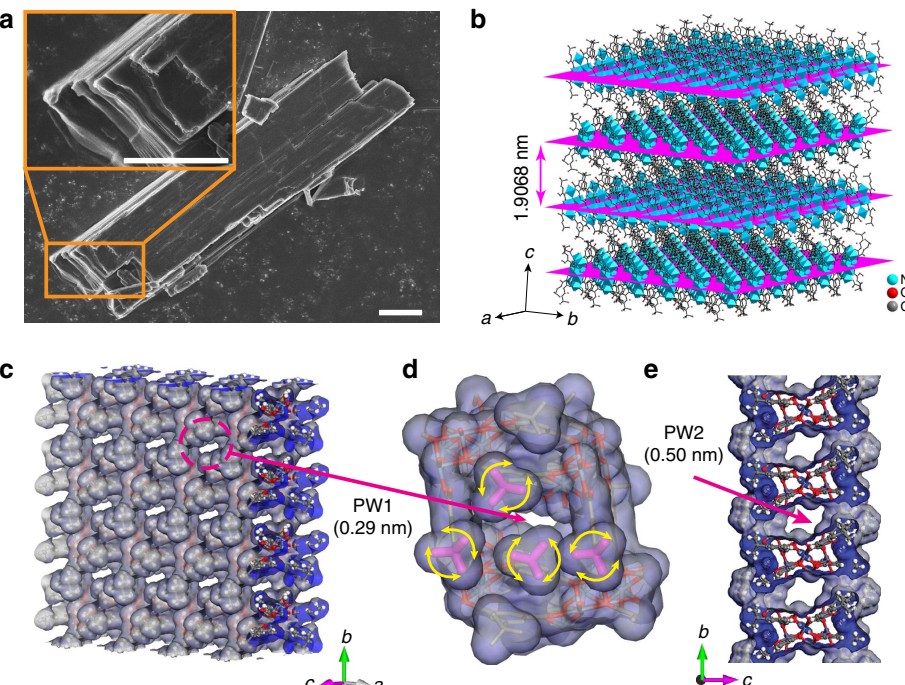

**Figure 1 | Layered structure and gas pathways of MAMS-1.** (**a**) FE-SEM image of layered MAMS-1 crystals. Scale bars, 5 μm. (**b**) Crystal structure of MAMS-1. The *ab* planes are highlighted in magenta to illustrate the layered structure. (**c**) Tilted vertical view of *ab* plane in MAMS-1 monolayer featuring PW1. (**d**) Amplified view of PW1 gated by two pairs of *tert*-butyl group highlighted in magenta. (**e**) View along *a* axis of MAMS-1 monolayer featuring PW2. The crystal structure is redrawn according to the single crystal structure of MAMS-1 (Cambridge Crystallographic Data Centre No. 617998) (ref. 70).

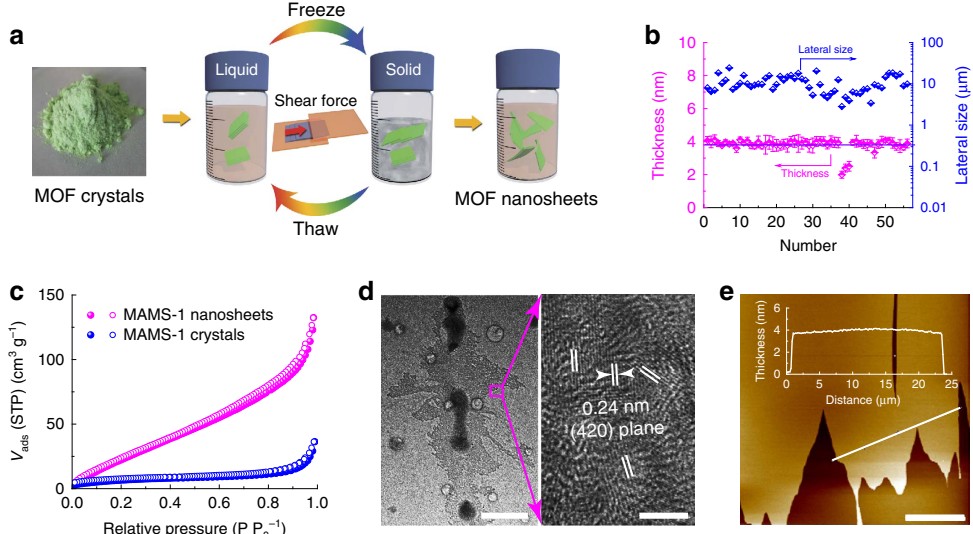

**Figure 2 | Exfoliation and purification of MAMS-1 nanosheets.** (**a**) The freeze-thaw exfoliation of MAMS-1 crystals into dispersed nanosheets. (**b**) Thickness and lateral size distribution of exfoliated MAMS-1 nanosheets after 10-cycle freeze-thaw in hexane (horizontal line indicates the theoretical thickness of a bilayered MAMS-1 nanosheet). (**c**) $N_2$ sorption isotherms at 77 K of MAMS-1 crystals and exfoliated MAMS-1 nanosheets (filled, adsorption; open, desorption). (**d**) TEM image of MAMS-1 nanosheets. Scale bars, 1 μm (left) and 3 nm (right). (**e**) AFM image of purified MAMS-1 nanosheets. Scale bar, 10 μm.

MAMS-1 crystals into 2D nanosheets based on the freeze-thaw process of solvents (Fig. 2a, Supplementary Fig. 1 and Supplementary Note 1). In a typical process, the MAMS-1 crystals are dispersed in hexane and frozen in liquid nitrogen bath ($-196\,^{\circ}$C) followed by thawing in hot water bath ($80\,^{\circ}$C). Recently, Zhu *et al.*[38] reported thermal-expansion-triggered gas exfoliation of bulk *h*-boron nitride based on their expansion and curling triggered by the huge temperature variation. Here, we propose that the shear force derived from the volumetric change of hexane between solid phase and liquid phase will be exerted on the suspended MAMS-1 crystals during the repeated freeze-thaw cycles, resulting in the exfoliation of MAMS-1 crystals into discrete nanosheets. As demonstrated by the atomic force microscopy (AFM) analyses on a total of 56 sites, more than 95% of them have a thickness of ca. 4 nm, proving the efficient exfoliation of MAMS-1 crystals into bilayered MAMS-1 nanosheets. However, the lateral size distribution of exfoliated MAMS-1 nanosheets is rather broad (4.7–24.3 μm, with an averaged value of 10.7 ± 4.8 μm, Fig. 2b, Supplementary Fig. 2 and Supplementary Note 2), which may not be suitable for membrane fabrication, and further purification is needed (*vide infra*). The efficient exfoliation of MAMS-1 can be further proven by the enhanced Brunauer-Emmett-Teller (BET) specific surface area of the exfoliated MAMS-1 nanosheets (from 24.8 m$^2$ g$^{-1}$ of bulk crystals to 126.1 m$^2$ g$^{-1}$ of exfoliated nanosheets, Fig. 2c). Especially, the external surface area ($S_{ext}$) of the exfoliated nanosheets increases by six times compared to the bulk crystals (Supplementary Table 1). Meanwhile, the micropore surface area ($S_{mic}$) also increases because of the exfoliation that makes some of the micropores in the bulk crystals accessible. Transmission electron microscopy (TEM) images of the exfoliated MAMS-1 nanosheets indicate a lattice spacing of $d = 0.24$ nm corresponding to the (420) crystal planes of MAMS-1 crystals (Fig. 2d). Fourier transform infrared (FTIR) spectra of the bulk MAMS-1 crystals and the exfoliated nanosheets are identical, indicating the preservation of MAMS-1 chemical structure during exfoliation (Supplementary Fig. 3). In addition, the exfoliated MAMS-1 nanosheets are thermally stable up to 300 °C, which is almost the same as the bulk crystals and therefore grants the application under high temperatures (Supplementary Fig. 4). All the above results have clearly demonstrated the effectiveness of freeze-thaw approach in exfoliating bulk MAMS-1 crystals into discrete MAMS-1 nanosheets with high aspect ratios (2,800 ± 1,200, calculated by the data in Fig. 2b), which is the key step towards the fabrication of nm-thick 2D molecular sieving membranes.

**Size-fractionation of MAMS-1 nanosheets.** A clean and transparent hexane suspension containing exfoliated MAMS-1 nanosheets was obtained by removing the un-exfoliated or aggregated particles after centrifugation at 10,000 r.p.m. for 20 min. Tyndall effect can be clearly observed from the colloidal suspension of MAMS-1 nanosheets (Supplementary Fig. 5), which is consistent with the reported colloidal suspensions of MOF or ZIF nanosheets[14,24,25]. However, fragmented nanosheets and nanoparticles can still be easily found from the suspension purified by centrifugation (Supplementary Figs 1,2). This may prevent the smooth and effective alignment of the exfoliated MAMS-1 nanosheets during membrane fabrication and lead to deteriorated separation performance[37]. Previously, pH-assisted selective sedimentation has been reported for the purification of GO nanosheets[39,40]. However, this method is not suitable for MOF and ZIF nanosheets due to their structural sensitivity toward pH values[41]. Recently, Coleman and co-workers revealed that good solvents for graphene dispersion should have nonzero polar and H-bonding Hansen parameters[42]. Inspired by this, we herein propose a solvent-selective sedimentation approach for the size fractionation of exfoliated MAMS-1 nanosheets. Briefly, hexane containing dispersed MAMS-1 nanosheets was layered on top of another immiscible solvent (for example, dimethylsulfoxide (DMSO) or N,N-dimethylformamide (DMF), Supplementary Fig. 6a). Given enough time, large MAMS-1 nanosheets would gradually sedimentate from the top hexane layer into the bottom layer under static conditions (Supplementary Note 3). After 2 weeks of sedimentation, ultra-large MAMS-1 nanosheets with a lateral size of more than 20 μm were obtained from the underlying solvent layer (Fig. 2e and Supplementary Fig. 6d).

In contrast, the top hexane layer contained mainly small nanosheets and nanoparticles (Supplementary Fig. 6b,c). Excitingly, the isolated ultra-large MAMS-1 nanosheets demonstrate excellent dispersion stability in DMSO suspensions without agglomeration or decomposition for longer than 4 months (Supplementary Fig. 6e), which largely facilitates the membrane fabrication.

**Fabrication of 2D MAMS-1 membranes.** In the previous studies of laminar membranes composed of 2D nanosheets, vacuum filtration is usually used to align the building blocks (for example, graphene or GO nanosheets) into membrane layers[5,6,43–45]. Alternatively, Yang and co-workers suggested a hot-drop casting method to fabricate nm-thick 2D ZIF membranes with both enhanced gas permeance and selectivity[14]. We herein use a similar hot-drop casting approach for membrane fabrication by aligning large MAMS-1 nanosheets onto porous substrates (anodic aluminium oxide, AAO, Whatman, 200 nm). 2D MAMS-1 membranes with different thicknesses (4-nm membrane, 12-nm membrane and 40-nm membrane) could be prepared simply by varying the volume of DMSO suspension used during hot-drop casting. The thinnest membrane (4-nm membrane) was obtained when 0.5 ml of DMSO suspension was used. However, MAMS-1 nanosheets were found to be sparsely distributed on the AAO substrate without forming a continuous membrane layer (Fig. 3a). To our surprise, many pinholes can be found within the nanosheets from the magnified field-emission scanning electron microscopy (FE-SEM) image (insert of Fig. 3a). Previously, Zhang and co-workers found that solvent evaporation from surrounding area could induce the flattening of graphene nanosheets on impermeable substrates[46]. In this study where porous AAO substrates were used, the solvent could evaporate and diffuse through the porous channels of the substrate. This process can exert a perpendicular capillary force to pull the fragile MAMS-1 nanosheets firmly towards the coarse AAO substrate causing rupture and pinholes of MAMS-1 nanosheets. Our speculation was further confirmed by the focused ion beam (FIB) TEM image of the cross-sectional area of the survived 4-nm membrane (Fig. 3d), wherein a thin layer (ca. 4 nm) can be found concaving towards the porous channels of AAO substrate due to the capillary force[26]. Given this condition, increasing the thickness of membrane layer should be able to preserve its integrity because the following MAMS-1 nanosheets covering on top of the first layer should experience less capillary force and thus may have a higher chance to survive. This hypothesis was

confirmed in a thicker membrane wherein 4.5 ml of DMSO suspension was used. A membrane thickness of around 12 nm was identified by the FIB-TEM image of the cross-sectional area (Fig. 3e). The continuous membrane layer is so thin that the Al element of underlying AAO substrate can be clearly detected by X-ray photoelectron spectroscopy (XPS, Supplementary Fig. 7 and Supplementary Note 4) and the porous texture of the underlying AAO substrate is also distinguishable (Fig. 3b). Different from the fragile MAMS-1 crystals, the MAMS-1 nanosheet layer is very flexible so that it can fold around the fractured AAO substrate (Supplementary Fig. 8 and Supplementary Note 5). The thickest membrane was obtained using 20 ml of DMSO suspension. Some crumples were observed from the membrane surface (Fig. 3c), which might be caused by the slow evaporation of DMSO. As shown in the FIB-TEM image (Fig. 3f), the membrane has a thickness of 40 nm indicating approximate ten layers of bilayered MAMS-1 nanosheets stacking together. Ni element can be clearly detected by energy-dispersive X-ray spectroscopy of the cross-section area of the membrane layer, confirming the expected elemental composition of MAMS-1 (insert of Fig. 3f). Powder X-ray diffraction (PXRD, Supplementary Fig. 9 and Supplementary Note 6) pattern of the 40-nm membrane indicates a peak from the (002) crystal plane of MAMS-1 (basal plane) while the peaks from the other crystal planes are undetectable, suggesting an alignment of MAMS-1 nanosheets along the [001] direction by which the PW1 with small aperture (ca. 0.29 nm) can be fully exposed.

**Gas separation performance of 2D MAMS-1 membranes.** Gas separation performance of the membranes was evaluated using an equimolar $H_2/CO_2$ mixture in a Wicke-Kallenbach permeation cell (Supplementary Fig. 10) at room temperature. The $H_2/CO_2$ separation factor of the 4-nm membrane (ca. 3, Supplementary Table 2) is smaller than the Knudsen diffusion selectivity ($\sqrt{M_{CO_2}/M_{H_2}} = 4.7$), indicating the existence of viscous flow matching well with the FE-SEM result. The 12-nm membranes exhibit $H_2/CO_2$ separation factors of $34 \pm 5$ and $H_2$ permeance of $6,516 \pm 990$ gas permeation units (GPU, 1 GPU = $3.3928 \times 10^{-10}$ mol m$^{-2}$ s$^{-1}$ Pa$^{-1}$, Supplementary Table 2). Interestingly, the gas permeance is almost three times higher than that of the 2D ZIF membranes ($2,280 \pm 490$ GPU with separation factors of $230 \pm 39$) (ref. 14), albeit with lower separation factors. Compared to the reported ZIF nanosheets, MAMS-1 nanosheets possess an extra gas pathway PW2 (Fig. 1e), which allows fast redistribution of the permeate gas molecules

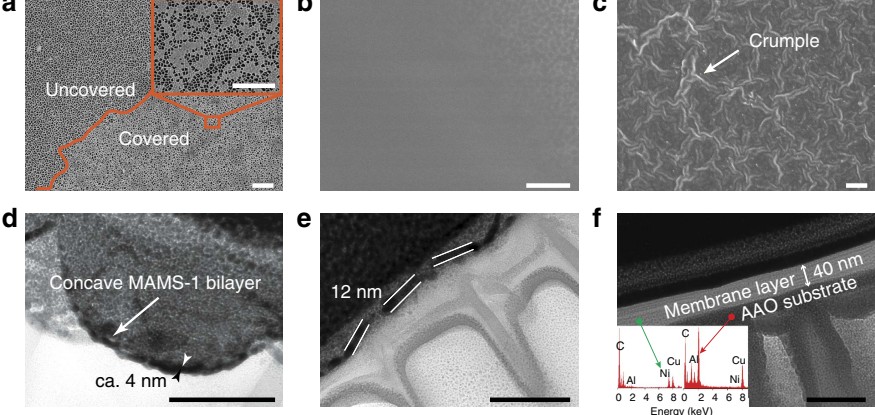

**Figure 3 | Morphology of 2D MAMS-1 membranes.** (a–c) FE-SEM images (top view) of 4-nm, 12-nm and 40-nm membranes, respectively. Scale bars, 2 μm. (d–f) FIB-TEM image (cross section) of 4-nm, 12-nm and 40-nm membranes, respectively. Scale bars, 100 nm. The dark layer on top of the coating is FIB-deposited platinum. Inset in **f**: energy-dispersive X-ray spectra of cross-sectional areas of MAMS-1 nanosheet layer (left) and AAO substrate (right).

leading to larger concentration gradient and thus enhanced driving force for gas permeation. When the membrane thickness is increased, the selectivity (separation efficiency) will be improved due to increased tortuosity and gas travel distance[47], but gas permeance (separation throughput) will be reduced. This can be confirmed in the 40-nm membranes, which possess $H_2/CO_2$ separation factors of $235 \pm 14$ but with compromised $H_2$ permeance of $553 \pm 228$ GPU (Supplementary Table 2).

The gas permselectivity and transport behaviour of the 40-nm membrane were further investigated by measuring the single gas permeation using He (kinetic diameter 0.255 nm), $H_2$ (0.289 nm), $CO_2$ (0.33 nm), $O_2$ (0.346 nm), $N_2$ (0.36 nm), $CH_4$ (0.38 nm) and $SF_6$ (0.513 nm). We observed a sharp cut-off of the permeance between small gases (He and $H_2$) versus large ones (Fig. 4a), indicating a clear molecular sieving gas separation performance. The permeation of $SF_6$ is too small to be detected by gas chromatograph detector. The permselectivities were calculated to be 268 for $H_2/CO_2$, 96 for $H_2/O_2$, 123 for $H_2/N_2$ and 164 for $H_2/CH_4$, which are all much higher than the Knudsen diffusion selectivities (blue line inserted in Fig. 4a). It is worth noting that the $CO_2$ permeance is even lower than that of $O_2$, $N_2$ and $CH_4$, which does not exactly follow the order of their kinetic diameters. Previously, Skoulidas and Sholl demonstrated a gas transport diffusivity sequence of $H_2 > N_2 \approx CH_4 > CO_2$ in MOF-5 by molecular dynamics (MD) simulation, which should be attributed to the $CO_2$-philicity feature of this MOF (ref. 48). We expect a similar $CO_2$-philicity of MAMS-1 because of the highly polar internal surface of PW2 contributed by the hydrophilic octanickel $[Ni_8(\mu_3-OH)_4]$ clusters[31], which is confirmed by the gas sorption isotherms (Supplementary Fig. 11 and Supplementary Note 7) and different adsorption heat between $CO_2$ ($23.0–31.2$ kJ mol$^{-1}$) and $H_2$ ($3.3–6.5$ kJ mol$^{-1}$) (Supplementary Fig. 12). Therefore, $CO_2$ molecules with a higher quadruple moment ($4.3 \times 10^{26}$ esu cm$^2$) should be trapped more strongly within PW2 than the other gases, leading to diffusion-controlled permeation along PW2 which is unfavourable for $CO_2$ (ref. 49).

Considering practical applications, we further evaluated the gas separation performance and long-term stability of the 40-nm

membrane for the separation of equimolar $H_2/CO_2$ mixtures. $H_2$ permeance decreased from 800 GPU for pure $H_2$ to 715 GPU for the mixed $H_2/CO_2$, with a separation factor of 245 (Supplementary Table 2). The decreased $H_2$ permeance in mixture is attributed to the partially hindered transport of $H_2$ molecules by the strongly adsorbed $CO_2$ molecules in PW2 due to the single-file diffusion in micropores[50,51]. Raising the test temperature to 40 °C caused an increased $H_2$ permeance to 880 GPU with a decreased separation factor of 225 (Supplementary Table 3). This is due to the increase of $CO_2$ diffusivity at elevated temperatures and hence agreeing well with the classical molecular sieving mechanism. The separation performance of the 40-nm membrane for $H_2/CO_2$ mixtures with various $H_2$ molar fractions was also evaluated (Supplementary Fig. 13 and Supplementary Note 8). $H_2$ permeance of 790 GPU and selectivity of 167 were obtained using a 20/80 $H_2/CO_2$ mixture (v/v), suggesting that the $H_2$ concentration of 20% in feed can be increased to as high as 97.66% in permeate simply by passing the mixture through the 40-nm membrane once. To the best of our knowledge, this result represents the highest separation performance of 2D membranes to separate low purity $H_2/CO_2$ mixtures (Supplementary Fig. 14 and Supplementary Table 3). Notably, although the thickness of the 40-nm membrane (ca. 40 nm) is much higher than that of the reported ultrathin 2D GO membranes (1.8-18 nm) (ref. 6), it still exhibits $H_2$ permeance almost three times higher than that of the GO membranes, possibly because of the shortened gas transport pathway contributed by the porous MAMS-1 nanosheets. The 40-nm membrane was continuously evaluated for the separation of equimolar $H_2/CO_2$ mixture up to 10,000 min without noticeable performance loss (Fig. 4b, Supplementary Fig. 15 and Supplementary Note 9), indicating its excellent stability for long-term continuous operations at room temperature.

**Gas permeation mechanism.** In order to elucidate the gas permeation mechanism, we conducted MD simulations to investigate the permeation behaviour of $H_2$ and $CO_2$ molecules through 2D MAMS-1 membranes, wherein the feed chamber and the

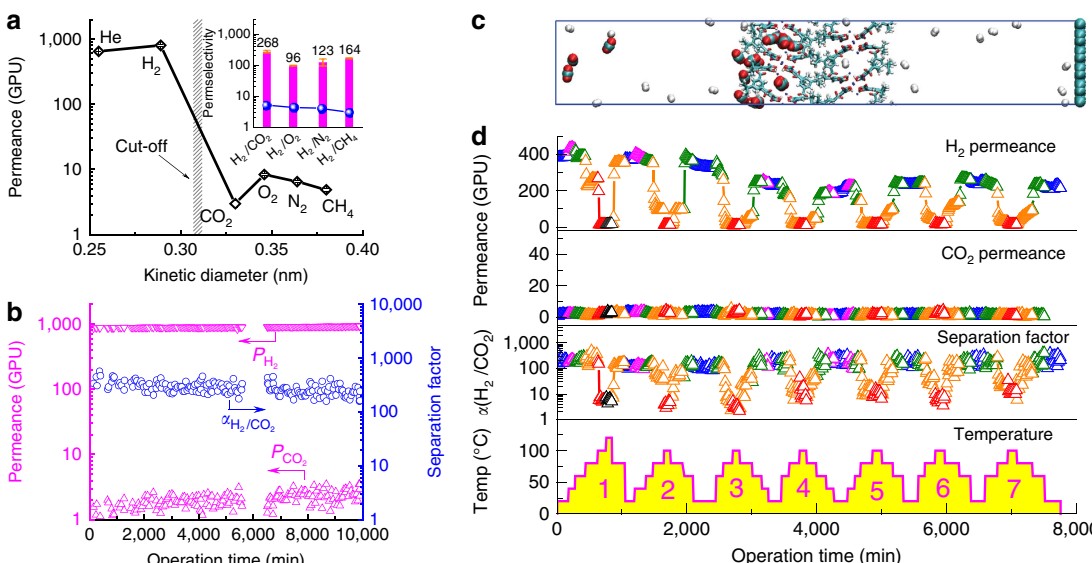

**Figure 4 | Single gas permeation and mixed gas separation performance of 2D MAMS-1 membranes.** (**a**) Single gas permeation of the 40-nm membrane (blue line in insert figure indicates the Knudsen diffusion selectivity of $H_2$ over other gases). (**b**) A 10,000 min continuous test of the 40-nm membrane for the separation of equimolar $H_2/CO_2$ mixture at room temperature. (**c**) A snapshot of MD simulation for the separation of equimolar $H_2/CO_2$ mixture through a bilayered MAMS-1 membrane (after 80 ns of simulation). (**d**) Gas permeance and $H_2/CO_2$ separation factors of the 40-nm membrane under seven heating/cooling cycles. Different colours represent various temperatures: blue, 20 °C; magenta, 40 °C; olive, 60 °C; orange, 80 °C; red, 100 °C and black, 120 °C.

permeate chamber were separated by a bilayered MAMS-1 nanosheet. In the case of single gas permeation simulation, no $CO_2$ molecule could permeate through the bilayered nanosheet within 80 ns of the whole simulation duration (Supplementary Fig. 16 and Supplementary Note 10). In contrast, 35% of $H_2$ molecules permeated through the bilayered nanosheet under the same condition (Supplementary Fig. 17 and Supplementary Note 11). Similar phenomenon was also observed for the simulation using an equimolar $H_2/CO_2$ mixture: 45% of $H_2$ molecules could permeate through the bilayered nanosheet, while the $CO_2$ molecules could only penetrate into PW2 of the first layer and then became trapped there (Fig. 4c, Supplementary Fig. 18, Supplementary Movie 1 and Supplementary Note 12). The simulation results strongly support our previous conclusion that low $CO_2$ permeance is attributed by the molecular sieving effects of the narrow PW1 aperture as well as the retarded diffusivity through PW2 in MAMS-1 nanosheets.

**Reversed thermo-switchability of 2D MAMS-1 membranes.** MAMS-1 has been reported with a thermo-responsive adsorption property, featuring enlarged gate openings at elevated temperatures (between $-196$ and $22\,°C$) that can be attributed to the intensified thermal vibration of the *tert*-butyl groups[31]. Therefore, we hypothesized that the 2D MAMS-1 membranes should demonstrate a similar thermo-responsive property by which the aperture of PW1 may become wider at elevated temperatures allowing the permeation of larger gas molecules. In order to test this hypothesis, another 40-nm membrane was measured using an equimolar $H_2/CO_2$ mixture at a temperature range of 20–100 °C (20–120 °C for the first cycle). The experiment was conducted by gradually increasing the test temperature to the set values (40, 60, 80 and 100 °C) at a heating rate of $2\,°C\,min^{-1}$ and keeping at each set temperature value until equilibrium of the separation performance was established. As has been expected, an increase of $H_2$ permeance from 392 to 430 GPU was observed when the temperature was initially increased from 20 to 40 °C (Fig. 4d). When the temperature was further increased to 60 °C, however, the $H_2$ permeance unexpectedly dropped back to approximate 390 GPU. Keeping increasing the temperature to 80 °C caused a further decrease of $H_2$ permeance to 256 GPU which is only 65.3% of the original value at 20 °C. When the test temperature was finally set at 100 and 120 °C, the membrane became almost impermeable to

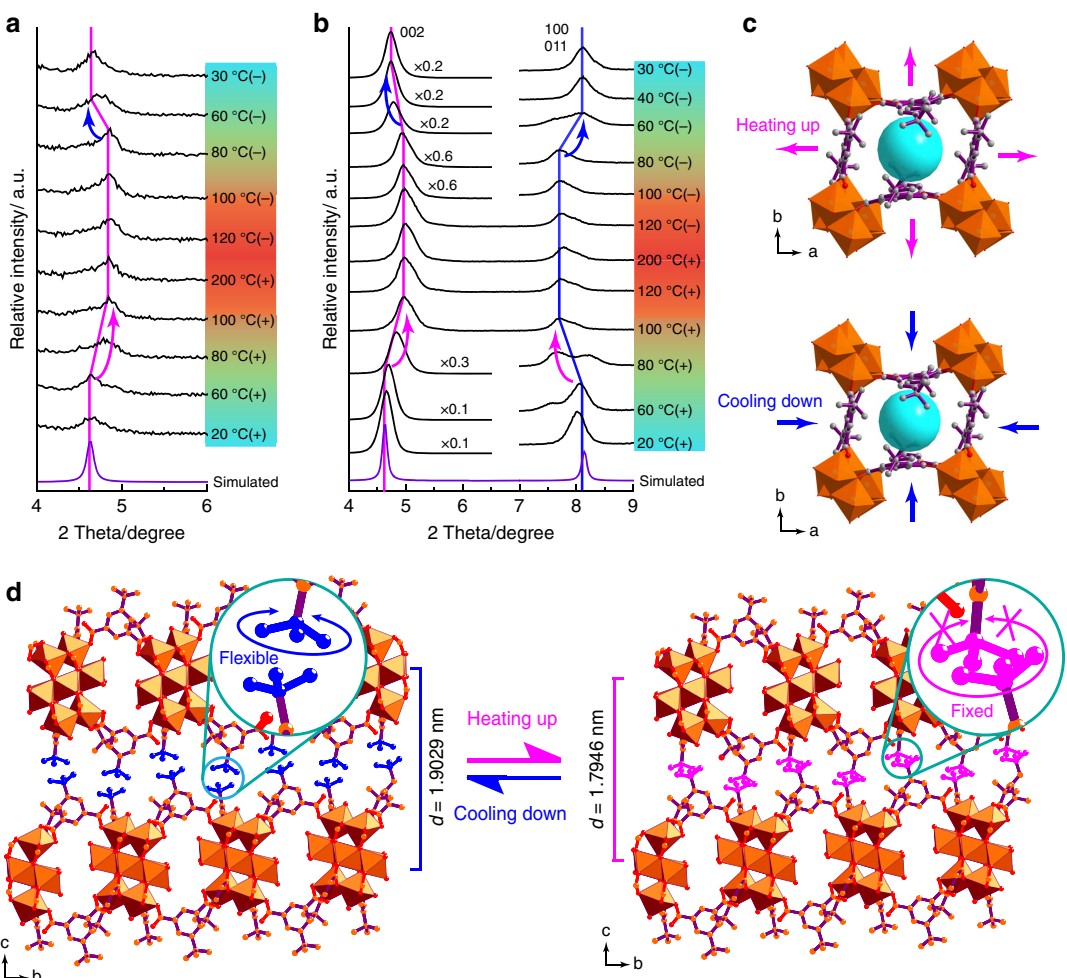

**Figure 5 | Structural flexibility of 2D MAMS-1 membrane.** (**a**) PXRD patterns of the 40-nm 2D MAMS-1 membrane under various temperatures. (**b**) PXRD patterns of the bulk MAMS-1 crystals under various temperatures. The violet line indicates the simulated PXRD pattern derived from the single crystal structure of MAMS-1 (Cambridge Crystallographic Data Centre No. 617998) (ref. 70). The symbol '$+$' represents heating up stage and the symbol '$-$' represents cooling down stage. (**c**) Illustration of the expansion (during heating) and shrinkage (during cooling) in *ab* planes of MAMS-1. The magenta arrows indicate the expansion during heating and the blue arrows indicate the shrinkage during cooling. (**d**) Illustration of the shrinkage (during heating) and expansion (during cooling) of interlayer distance in MAMS-1. The freely rotated *tert*-butyl groups are highlighted in blue and the frozen ones are highlighted in magenta.

$H_2$ with a permeance of merely 14 GPU, which is only 3.6% of the original value at 20 °C. Surprisingly, $H_2$ permeance bounced back to 356 GPU when the test temperature was reduced from 120 to 80 °C. After cooling down to 20 °C, the $H_2$ permeance stabilized at 373 GPU which is close to the initial value (95.2%). This process was partially reversible with a fluctuation of the $H_2$ permeance at 20 °C, which stabilized at 230 GPU after seven heating/cooling cycles. Notably, $CO_2$ permeance rarely changed during the seven heating/cooling cycles (1–3 GPU). As a result, $H_2/CO_2$ separation factor also exhibited a temperature dependent behaviour, with the highest value of 245 at 20 °C and the lowest value of ca. 5 at 100 °C.

The above reversed thermal-switchable behaviour, which was also observed in other membranes of this study (Supplementary Fig. 19 and Supplementary Note 13), contradicts our hypothesis and cannot be explained by the classical transport theory for molecular sieving mechanism, wherein the gas permeance should increase at elevated temperatures following the Arrhenius equation[52,53]. We speculated that this abnormal behaviour should be contributed by the structural flexibility of the MAMS-1 nanosheets. In order to confirm this, the 40-nm membrane and bulk MAMS-1 crystals were further characterized by in situ variable temperature PXRD. Notably, the PXRD peak from the (002) crystal plane of the 40-nm membrane shifted towards higher two-theta angles from 4.64 to 4.92° upon heating from 20 to 100 °C (Fig. 5a). The peak remained almost unchanged upon further heating up to 200 °C, but started to shift back when the temperature was reduced below 80 °C and finally reached 4.64° at 30 °C. From 20 to 100 °C, a change of 0.28° towards higher two-theta angle was achieved in the (002) peak, indicating the contraction of lattice spacing between (002) planes from 1.9029 to 1.7946 nm based on the Bragg equation. A similar shift of the (002) peak upon heating/cooling was observed in the bulk MAMS-1 crystals (from 4.62° to 4.96° as shown in Fig. 5b). Interestingly, the (100) and (011) peaks of bulk MAMS-1 crystals shifted to 7.72° during heating, and returned back to 8.14° upon cooling, corresponding to an expansion of 0.0589 nm in the ab crystalline planes and enlarged aperture size of PW1 at elevated temperatures (Fig. 5c), which agrees well with the temperature-induced molecular-gating effects of MAMS-1 proposed by Zhou et al.[31] However, although the aperture size of PW1 becomes larger at higher temperatures, its kinetic opening is still controlled by the rotation of tert-butyl groups. Preliminary MD simulation indicates that the MAMS-1 nanosheet is even impermeable to $H_2$ molecules if the free rotation of tert-butyl groups is prohibited. The tert-butyl groups can rotate freely at room temperature because of the surrounding free volume (Fig. 5d). At higher temperatures, the free rotation of tert-butyl groups will be restricted due to the intensified steric hindrance caused by reduced interlayer distance, leading to blocked PW1 and sharply decreased gas permeance. Although several MOFs have been reported with stimuli-responsive features[31,34,54–56], this is the first time that reversed thermo-switchable molecular sieving is demonstrated in molecular sieving membranes composed of 2D MOF nanosheets[57], which may find novel applications in temperature-related gas separations.

## Discussion

We have prepared nm-thick 2D membranes composed of exfoliated MOF nanosheets demonstrating reversed thermo-switchable molecular sieving gas separation performance. In order to obtain large and defect-free MOF nanosheets as the building blocks for molecular sieving membranes, we pioneered the mild exfoliation of a layered MOF (MAMS-1) via freeze-thaw approach in suitable solvent systems. In addition, we proposed for the first time a solvent-selective sedimentation approach for size fractionation and collection of ultra-large MAMS-1 nanosheets with lateral size of more than 20 μm. The exfoliated and purified MAMS-1 nanosheets were well-aligned into membranes with various thicknesses, wherein the challenges were elucidated with offered solutions that will be helpful for the fabrication of other 2D MOF membranes. The membranes derived from well-aligned MAMS-1 nanosheets demonstrated an unprecedented reversed thermos-switchable $H_2$ permeation, which can be attributed to the structural flexibility of MAMS-1 nanosheets that has yet to be observed in other inorganic membranes. This work sheds light on the tailored synthesis of smart 2D membranes with wide applications in clean energy and environmental sustainability.

## Methods

**Synthesis of MAMS-1 crystals.** 5-tert-butyl-1,3-benzenedicarboxylic acid (5-bbdc, 1.5 g, 6.8 mmol) and $Ni(NO_3)_2 \cdot 6H_2O$ (3.0 g, 10.2 mmol) were suspended in 150 ml of 20 v% ethylene glycol aqueous solution. The mixture was placed in a 200-ml Teflon container, sealed in an autoclave and heated to 210 °C with a heating rate of 2 °C min$^{-1}$. After 24 h, the reaction was stopped by cooling down to room temperature and MAMS-1 was obtained as light-green needle-like crystals and washed with DMF five times to completely remove the residual ligand and metal salt. Before gas sorption tests, the crystals were further solvent-exchanged with methanol three times and dried under vacuum at 150 °C overnight.

**Exfoliation by sonication.** The hexane suspension containing MAMS-1 crystals with a concentration of 1.0 mg ml$^{-1}$ was treated in an ultrasonic bath (Elmasonic E 30 H, 240 W) for 30 min. A colloidal suspension of MAMS-1 nanosheets was obtained after centrifugation at 10,000 r.p.m. for 20 min (Dynamica Scientific Ltd., Velocity 14) to remove un-exfoliated particles. Further purification of the nanosheets was conducted by free-standing the colloidal suspension for at least 2 weeks.

**Exfoliation by freeze-thaw approach.** In a typical process, the MAMS-1 crystals were dispersed in various solvents (water, 20 v% ethanol aqueous solution, 80 v% ethanol aqueous solution, ethanol and hexane) with a concentration of 1.0 mg ml$^{-1}$ and heated in hot water bath (80 °C) for 3–5 min, and then immediately frozen in liquid nitrogen bath (−196 °C) until complete freeze. After that, the solidified mixtures were thawed in hot water bath (80 °C) again. The freeze-thaw cycle was repeated several times depending on the solvent. The un-exfoliated MAMS-1 crystals were removed from the supernatant by centrifugation at 10,000 r.p.m. for 20 min. In the case of hexane, the exfoliation rate was about 6.5% with a concentration of 0.065 mg ml$^{-1}$ for the suspension measured by weighting AAO substrates before and after drop-casting.

**Size-fractionation of MAMS-1 nanosheets.** The exfoliated MAMS-1 nanosheets in hexane suspension were size fractionated by solvent-selective sedimentation. Briefly, two immiscible solvents were vertically layered based on their different density such as a top layer of exfoliated MAMS-1 nanosheets in hexane suspension (ca. 20 ml) and a bottom layer of DMSO or DMF (ca. 1 ml). Sedimentation of large MAMS-1 nanosheets from top layer to bottom layer would occur naturally under static conditions. After at least 2 weeks, 50% of bottom layer was collected for further characterization and membrane fabrication.

**Scaled-up preparation of MAMS-1 nanosheet powder.** MAMS-1 nanosheet powder was obtained by freeze-drying. Briefly, 10 ml of p-xylene was added into 400 ml of MAMS-1 nanosheets in hexane suspension. The mixture was treated with a rotary evaporator at 30 °C under vacuum to remove hexane, and then with a freeze-dryer under high vacuum to remove frozen p-xylene affording MAMS-1 nanosheet powder, which was used for FTIR, thermogravimetric analysis (TGA) and gas sorption tests.

**Membrane fabrication.** 2D MAMS-1 membranes were fabricated by drop-casting various volumes of DMSO suspension (0.5 ml for 4-nm membrane, 4.5 ml for 12-nm membranes and 20 ml for 40-nm membranes) containing exfoliated and purified bilayered MAMS-1 nanosheets on AAO substrate. The temperature for drop-casting was kept at 200 °C by heating AAO substrate on a hot plate to facilitate the evaporation of DMSO.

**Gas permeation tests.** In order to avoid the damage of MAMS-1 nanosheet layer, the edge of the membrane disk was masked with a high temperature aluminium gasket coated with silicone rubber pad, exposing only 5-mm-diameter hole in the centre of the membrane. The volumetric flow rate of gas (either single gas or mixed gas) was kept at 50 ml min$^{-1}$ by mass flow controllers (Brooks Instrument).

Argon was used as the sweep gas at a constant volumetric flow rate of 50 ml min$^{-1}$ to eliminate concentration polarization in the permeate side. There was no pressure drop between the two sides of the membrane to prevent any distortion of the MAMS-1 nanosheet layer. The gas permeance ($P_i$, GPU) and permselectivity of hydrogen over other gases ($S_{H_2/i}$) were calculated by the following equations,

$$P_i = \frac{J_i}{3.3928 \times 10^{10} \Delta P_i} \tag{1}$$

$$S_{H_2/i} = \frac{J_{H_2}}{J_i} \tag{2}$$

where $J_i$ is the gas permeation flux through membrane, mol m$^{-2}$ s$^{-1}$; $\Delta P_i$ is the transmembrane pressure difference of component $i$, Pa. The separation factor $(\alpha_{H_2/CO_2})$ was defined as the molar ratio of $H_2$ to $CO_2$ in the permeate and feed side determined by gas chromatograph (Shimadzu GC-2014),

$$\alpha_{H_2/CO_2} = \frac{y_{H_2}/y_{CO_2}}{x_{H_2}/x_{CO_2}} \tag{3}$$

where $x_{H_2}$ and $x_{CO_2}$ are the molar fractions of $H_2$ and $CO_2$ in the feed, respectively; $y_{H_2}$ and $y_{CO_2}$ are the molar fractions of $H_2$ and $CO_2$ in the permeate, respectively. Each separation factor was calculated by the average of at least ten measurements. In order to avoid the possibility of gas sorption in silicone rubber pad, all the tests were conducted after the establishment of steady-state (for example, overnight equilibrium).

**Molecular dynamics simulation.** The simulation system was illustrated by a bilayered MAMS-1 nanosheet. There were two chambers containing pure gas or equimolar mixture of $H_2/CO_2$ (20 molecules for each component, Supplementary Figs 16a,17a and 18a) and a vacuum, respectively. A graphene plate was added on the right side of the system to separate the feed and permeate chambers. The periodic boundary conditions were applied in the $x$ and $y$ directions; thus the membrane was mimicked to be infinitely large on the $xy$ plane. To mimic the experimentally observed molecular-gating effect[31], the flexibility of the MAMS-1 nanosheet was incorporated. The parameters in bonded and nonbonded interactions were derived using OBGMX (ref. 58) on the basis of the universal force field[59]. A large number of simulation studies have shown that the universal force field can well predict the adsorption and diffusion of guests in various MOFs (refs 48,60–63). The bonded interactions include bond stretching, bending and torsional potentials,

$$U_{bonded} = U_{stretching} + U_{bending} + U_{torsional} \tag{4}$$

$$U_{stretching} = \sum \frac{1}{2} k_r \left( r_{ij} - r_{ij}^0 \right)^2 \tag{5}$$

$$U_{bending} = \sum \frac{1}{2} k_\theta \left( \theta_{ijk} - \theta_{ijk}^0 \right)^2 \tag{6}$$

$$U_{torsional} = \sum k_\phi \left[ 1 + \cos \left( m\,\phi_{ijkl} - \phi_{ijkl}^0 \right) \right] + \sum k_\xi \left[ 1 + \cos \left( m\,\xi_{ijkl} - \xi_{ijkl}^0 \right) \right] \tag{7}$$

where $k_r$, $k_\theta$, $k_\phi$ and $k_\xi$ are the force constants; $r_{ij}$, $\theta_{ijk}$, $\phi_{ijkl}$ and $\xi_{ijkl}$ are bond lengths and angles, proper and improper dihedrals, respectively; $m$ is the multiplicity and was set to two for most dihedrals; $r_{ij}^0$, $\theta_{ijk}^0$, $\phi_{ijkl}^0$ and $\xi_{ijkl}^0$ are the equilibrium values. The nonbonded interactions include Lennard-Jones (LJ) and electrostatic potentials,

$$U = \sum 4\varepsilon_{ij} \left[ \left( \frac{\sigma_{ij}}{r_{ij}} \right)^{12} - \left( \frac{\sigma_{ij}}{r_{ij}} \right)^6 \right] + \sum \frac{q_i q_j}{4\pi\varepsilon_0 r_{ij}} \tag{8}$$

where $\varepsilon_{ij}$ and $\sigma_{ij}$ are the well depth and collision diameter, $r_{ij}$ is the distance between atoms $i$ and $j$, $q_i$ is the atomic charge of atom $i$, and $\varepsilon_0 = 8.8542 \times 10^{-12}\,C^2\,N^{-1}\,m^{-2}$ is the permittivity of vacuum. The atomic charges were calculated using the extended charge equilibration method (EQeq) (ref. 64). $CO_2$ molecule was mimicked by the elementary physical model with the C–O bond length of 1.161 Å and the bond angle ∠OCO of 180° (ref. 65). $H_2$ was modelled by the consistent valence force field[66], which was shown to have good performance for $H_2$ storage in carbon nanotubes[67]. The carbon atoms in graphene plate were mimicked by LJ potential as used for carbon nanotubes[68].

The simulation system was initially subjected to energy minimization using the steepest descent method with a maximum step size of 0.1 Å and a force tolerance of 1 kJ mol$^{-1}$ Å$^{-1}$. Then, MD simulation was carried out at 300 K. The temperature was controlled by the velocity-rescaled Berendsen thermostat with a relaxation time of 0.1 ps. The MAMS-1 nanosheet was flexible during the simulation and a position restrain with a force constant of 10$^5$ was added to the Ni atoms. A cut-off of 10 Å was used to calculate the LJ interactions and the particle-mesh Ewald method was used to evaluate the electrostatic interactions with grid spacing of 1.2 Å and real-space cut-off of 10 Å. A time step of 2 fs was used to integrate the equations of motion by leapfrog algorithm. The simulation duration was 80 ns and the trajectory was saved every 4 ps. GROMACS version 4.5.3 was used to conduct the simulation[69].

**Characterization.** FTIR spectra were obtained with a Nicolet 6700 FTIR spectrometer. PXRD patterns were obtained on a Rigaku MiniFlex 600 X-ray powder diffractometer equipped with a Cu sealed tube ($\lambda = 1.54178$ Å) at a scan rate of 2° min$^{-1}$. TGA was performed using a Shimadzu DTG–60AH thermal analyser under a flowing air (20 ml min$^{-1}$) with a heating rate of 10 °C min$^{-1}$. FE-SEM was conducted on a JEOL JSM-7610F scanning electron microscope. Samples were treated via Pt sputtering for 100 s before observation. TEM was conducted on a JEOL JEM-3010 transmission electron microscope. Prior to FIB cutting, the MAMS-1 layer was sandwiched between the FIB-deposited platinum (to protect the coating from milling) and the AAO support. Then, the cross-section was observed by TEM. AFM was conducted by testing samples deposited on silica wafers using tapping mode with a Bruker Dimension Icon atomic force microscope.

XPS experiments were performed with a Kratos AXIS Ultra DLD surface analysis instrument using a monochromatic Al K$_\alpha$ radiation (1486.71 eV) at 15 kV as the excitation source. The takeoff angle of the emitted photoelectrons was 90° (the angle between the plane of sample surface and the entrance lens of the detector). Peak position was corrected by referencing the C 1s peak position of adventitious carbon for the sample (284.8 eV), and shifting all other peaks in the spectrum accordingly. Fitting was done using the program CasaXPS. Each relevant spectrum was fit to a Shirley/Linear type background to correct for the rising edge of backscattered electrons that shifts the baseline higher at high binding energies. Peaks were fit as asymmetric Gaussian/Lorentzians, with 0–30% Lorentzian character. The FWHM of all sub-peaks was constrained to 0.7–2 eV, as dictated by instrumental parameters, lifetime broadening factors and broadening due to sample charging. With this native resolution set, peaks were added, and the best fit, using a least-squares fitting routine, was obtained while adhering to the constraints mentioned above.

Gas sorption isotherms of MAMS-1 crystals and MAMS-1 nanosheet powder were measured using a Micromeritics ASAP 2020 surface area and pore size analyser. Before the measurements, the samples were degassed under high vacuum (<0.01 Pa) at 150 °C for 10 h. UHP grade $H_2$, $CO_2$ and $N_2$ were used for all the measurements. Oil-free vacuum pump and oil-free pressure regulators were used to prevent contamination of the samples during the degassing process and isotherm measurement. The temperatures of 77, 273 and 298 K were maintained with a liquid nitrogen bath, an ice water bath and under room temperature, respectively. The Brunauer-Emmett-Teller equation was used to calculate the specific surface area from adsorption data obtained at $P/P_0 = 0.05 - 0.3$. The external surface area was calculated by the $t$-plot method with Halsey equation.

**Data availability.** The authors declare that all the data supporting the findings of this study are available within the article (and Supplementary Information Files), or available from the corresponding author on reasonable request.

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

## Acknowledgements

This work is supported by National University of Singapore (CENGas R-261-508-001-646 and R-279-000-474-112) and Singapore Ministry of Education (MOE AcRF Tier 2 R-279-000-429-112). We thank Prof. Shaoming Ying for the help in crystallography.

## Author contributions

D.Z. formulated and supervised the project. X.W. prepared MAMS-1 nanosheets, fabricated membranes and performed TGA, PXRD, FTIR, FIB-TEM, gas sorption and gas permeation tests. C.C. collected the FE-SEM images of membranes. K.Z., K.M.G. and J.J. constructed the molecular models and conducted the MD simulations. Y.Q. collected the TEM images of MAMS-1 nanosheets. Z.K. built the gas permeation cell. X.W. and D.Z. wrote the paper, and all authors contributed to revising the paper.

## Additional information

**Competing financial interests:** The authors declare no competing financial interests.

