## [Peer Review File · Nature Communications]

Reviewers' Comments:

Reviewer #1 (Remarks to the Author)

The manuscript describes synthesis/characterization of MOF nanosheets based on a top-down exfoliation method and gas (H₂, CO₂) permeability. The research subject is important. On the other hand, the idea and the method for preparing the nanosheets are not new, and all analysis and results including the gas permeability results are not specifically great (expected level), either. In addition, the thermal-switchable behavior has been reported in the bulk crystal. Therefore, it is not surprising that the exfoliated nanosheets show the similar behavior. Therefore, I do not recommend the present manuscript to be published Nature Communication. Detailed comments are listed as follows.

1) The main text does not have to contain the details of solvent selection for the exfoliation as it is not the special case that you consider appropriate solvent depending on the materials. For example, sentences as follows "When the same solvent was used in this study, however, no exfoliated MAMS-1 nanosheet could be obtained even after 20 cycles of freeze-thaw (Figure S2a)..... 26;.". can be move to Methods or Supplementary Information.

2) The author should provide more details how to calculate the BET surface area as the BET value is strongly affected by the arbitrarily selected points. Does the BET surface represent the regulated nanopores or surface of the objects? From the shape of the N₂ sorption isotherm of the nanosheet, which does not have any plateaus, I suspect that the nanosheets are not uniform, but include diverse size-scaled objects. If the nanosheets are uniform and thickness are less than 10 nm, uptake of N₂ should increase much sharply in the high P region ($P/P_0 = 0.8 - 1.0$) as shown in ref 13 and 21.

3) Regarding the size of the exfoliated nanosheets, the authors should provide statistical analysis on both the thickness and the lateral sheet size.

4) In order to discuss the pathway of the prepared samples for gas permeability, the author should demonstrate the XRD of the nanosheets on AAO substrates.

5) Gas permeability results generally depend strongly on how the nanosheets sample prepared on the porous substrates and it is no possible to prepare the identical samples. The authors prepared three different types of sample (M1, M2 and M3). But these categorization is ambiguous. The authors should evaluate the error on the gas selectivity and transparency in each categorized sample.

6) In order to understand the reversed thermal-switchable behavior seen in the nanosheets, the authors examined temperature dependence of PXRD on bulk crystals. It often seen that structural/phase change behavior is different between bulk materials and nanoscaled materials. The authors should demonstrate the PXRD measurements on exfoliated nanosheets.

Reviewer #2 (Remarks to the Author)

This manuscript describes the fabrication of nanometer-thick 2D membranes displaying thermo-switchable molecular sieving CO₂/H₂ separation performance. The content is an interesting contribution where the two hot and timely concepts of 2D nanosheets and membrane-based gas separation are linked. However, there are issues here which would prevent the publication of this work in Nat. Comm.. The authors reported the synthesis of bilayer-nanosheets from a MOF crystal and confirmed the structural features like porosity and thickness through several characterization methods. However, they did not make direct use of these bilayer-nanosheets for CO₂/H₂

separation because of the difficulties in the membrane fabrication process as they mentioned. Instead, they make the membrane with the thickness of 40 nm. Is the porosity of this 40 nm-thick membrane (M3-1) the same as that of the bilayer-4 nm-membrane? If not, what is the structure of these 40 nm-thick material on the AAO-substrate? Are the exfoliated bilayer nanosheets stacked again? Fig 3c and f may be not sufficient. The structural features of these 40-nm materials must be clarified since they are not bilayer-nanosheets (4 nm). The authors need to provide more information for these 40 nm-material besides XRD. M3-1 for CO₂ separation (page 10, line 207 and line 222): The authors cannot use bulky MOF crystals to explain the separation on your nanosheets. Please measure the CO₂/H₂ gas uptake performance for the target-membrane (M3-1), like what was reported in Ref. 21, to get a better understanding of these significant separation differences. In addition, are all exfoliated nanosheets two layers (4 nm)? Or this is average number? How about the XRD structure of the 2-layers nanosheets? The N₂-adsorption curves look unmoral in comparison with that of ZIF-sheets (Ref. 13) and CuBDC-MOF-sheets (Ref. 21). Why? What is pore size distribution of these Nanosheets? This is important to understand the gas adsorption and diffusion. For solvent-selective sedimentation approach, the authors suggest that the high dipole moment of DMSO is a key factor. More experiments with different solvents are needed to confirm this suggestion. Page 8 line 164, what is thickness of M2 membranes? Reference section: Pay attention on the abbreviation (several typos)

Reviewer #3 (Remarks to the Author)

This work reported an ultrathin bilayered MAMS-1 membrane for the gas separation of H₂/CO₂ pair. The exfoliated MAMS-1 nanosheets were purified by a novel selective sedimentation approach, which may give the hints for other researchers worked on 2D membranes. The thermo-responsive property of parent MOF materials gives the obtained 2D membrane a thermal-switchable behavior on the gas separation. This gating effect of membrane materials may open the window of multi-function membrane separation. This is a state-of-the-art work. It will be a significant contribution to this field and will be of great interest to a broad readership of Nature Communications. I have several minor comments before its publication.

1. As the author mentioned, larger pieces of MAMS-1 layer were obtained in this work, so how about the mechanical strength?
2. The purification of the 2D nanosheets is interesting. How about the diameter distribution of the nanosheets, and a larger scale AFM image contained several pieces of MAMS-1 layers should be better. And if the relationship between the purification yield and time could be studied and supplied, it will be very help to understand the separation process of this method.
3. The single gas test part should be moved to the ahead of the first paragraph of mixture gas test part to let the manuscript flow. In the scheme picture of gas test setup (Figure S9), there will be a gas path of Ar connected to GC, if the TCD was used as detector.
4. It will be better to sum up more information of all the membranes in Table S1, such as membrane thickness and gas test condition.
5. There is a mistake in Table S2. The performance of JUC-150 listed in the table is not right. (It is the data of another membrane with a longer pillar of 4,4-bipyridine one).

Reviewer 1

The manuscript describes synthesis/characterization of MOF nanosheets based on a topdown exfoliation method and gas (H₂, CO₂) permeability. The research subject is important. On the other hand, the idea and the method for preparing the nanosheets are not new, and all analysis and results including the gas permeability results are not specifically great (expected level), either. In addition, the thermal-switchable behavior has been reported in the bulk crystal. Therefore, it is not surprising that the exfoliated nanosheets show the similar behavior. Therefore, I do not recommend the present manuscript to be published Nature Communication. Detailed comments are listed as follows.

Response: We thank the reviewer for this comment. In the following statement, we will further emphasize the innovations of this work. **(1) This is the first report regarding the exfoliation of laminar MOF crystals using a mild freeze-thaw approach.** Currently, MOF nanosheets are usually obtained by peeling off from the bulk crystals through ball-milling or sonication. However, the integrity of MAMS-1 nanosheets in this study cannot be preserved during sonication-based exfoliation (Figure S2g). This can be attributed to the poor mechanical strength of MAMS-1 nanosheets, which has been further proven by the nanosheet fracture during membrane fabrication (Figure 3a and Figure 3d). Given the above concerns, we adopted a mild freeze-thaw approach in this study to exfoliate MAMS-1 with the maximum preservation of the resultant nanosheets. To the best of my knowledge, this is the first report regarding the exfoliation of laminar MOF crystals using freeze-thaw approach, and should be of great interest to the field of 2D materials. **(2) This is the first report regarding the reversed thermo-switchable molecular sieving effect of nm-thick 2D MOF membranes.** Just as mentioned by the reviewer, 2D membrane for gas separation is an interesting and promising topic. However, it is extremely difficult to fabricate nm-thick 2D membranes with high gas separation performance. In addition, the thermosensitive behavior of bulk MOF crystals might change once the exfoliated nanosheets are stacked to

form a membrane layer (Figure 5a-c). In this work, we find the 2D MAMS-1 membrane is almost impermeable to hydrogen at high temperatures (100 °C and above), which does not fully agree with the originally reported thermosensitive behavior of bulk MAMS-1 crystals (enlarged aperture size at elevated temperature, *Angew. Chem., Int. Ed.* **2007**, *46*, 2458). We have found that the gas permeance of 2D MAMS-1 membrane is not only determined by the aperture size of nanosheets but also the interlayer spacing between them. This is an important contribution to the field of membrane, which should be helpful for the design and screen of novel 2D MOF nanosheets as the building blocks for smart membranes. Detailed responses are given below and the according revisions have been made in the revised manuscript.

(1) The main text does not have to contain the details of solvent selection for the exfoliation as it is not the special case that you consider appropriate solvent depending on the materials. For example, sentences as follows “When the same solvent was used in this study, however, no exfoliated MAMS-1 nanosheet could be obtained even after 20 cycles of freeze-thaw (Figure S2a).....”. can be move to Methods or Supplementary Information.

Response: We thank the reviewer for this comment. As suggested, the details of solvent selection for the exfoliation have been moved to Supplementary Information (Figure S2 section). Meanwhile, we have also simplified the description regarding other experimental details in the main text.

(2) The author should provide more details how to calculate the BET surface area as the BET value is strongly affected by the arbitrarily selected points. Does the BET surface represent the regulated nanopores or surface of the objects? From the shape of the N₂ sorption isotherm of the nanosheet, which does not have any plateaus, I suspect that the nanosheets are not uniform, but include diverse size-scaled objects. If the nanosheets are

uniform and thickness are less than 10 nm, uptake of N₂ should increase much sharply in the high P region (P/P₀ = 0.8 – 1.0) as shown in ref 13 and 21.

Response: We thank the reviewer for this comment. More details on how to calculate the BET surface area have been added in the revised Supplementary Information as follows: “The BET equation was used to calculate the specific surface area from adsorption data obtained at $p/p_0 = 0.05-0.3$. The external surface area was calculated by the t-plot method with Halsey equation.” The data points used for BET surface area calculation are shown in **Figure R1**. The Y-intercept is positive and the plot of $1/[Q(P_0/P)-1]$ vs P/P_0 clearly falls on a straight line. The increased BET surface area can be attributed to more exposed micropores and external surface area of exfoliated MAMS-1 nanosheets, which is consistent with the conclusion in ref 13 and 21. Thus, we have revised the discussion of BET surface area in the main text as follows: “The efficient exfoliation of MAMS-1 can be further proven by the enhanced Brunauer-Emmett-Teller (BET) specific surface area of the exfoliated MAMS-1 nanosheets (from $24.8 \text{ m}^2 \text{ g}^{-1}$ of bulk crystals to $126.1 \text{ m}^2 \text{ g}^{-1}$ of exfoliated nanosheets, Figure 2c and Table S1). Especially, the external surface area (S_{ext}) of the exfoliated nanosheets increases by 6 times compared to the bulk crystals. Meanwhile, the micropore surface area (S_{mic}) also increases because of the exfoliation that makes some of the micropores in the bulk crystals accessible.” A statistical analysis of the lateral size and thickness of the exfoliated nanosheets was conducted and shown in Figure 2b and Figure S3. As suggested by the reviewer, the lateral size distribution of exfoliated MAMS-1 nanosheets is indeed broad (ranging from $4.7 \text{ }\mu\text{m}$ to $24.3 \text{ }\mu\text{m}$), which should be responsible for the sharply increased uptake of N₂ at high pressure region in the isotherms. The according revision has been added into the main text as follows: “As demonstrated by the atomic force microscopy (AFM) analyses on a total of 56 sites, more than 95 % of them have a thickness of *ca.* 4 nm, proving the efficient exfoliation of MAMS-1 crystals into bilayered MAMS-1 nanosheets. However, the lateral size distribution of exfoliated MAMS-1 nanosheets is rather broad (4.7

- 24.3 μm , with an averaged value of $10.7 \pm 4.8 \mu\text{m}$, Figure 2b and Figure S3), which may not be suitable for membrane fabrication and further purification is needed (*vide infra*)."

Figure R1. BET plots for MAMS-1 crystals and MAMS-1 nanosheets.

(3) Regarding the size of the exfoliated nanosheets, the authors should provide statistical analysis on both the thickness and the lateral sheet size.

Response: We thank the reviewer for this comment. A statistical analysis of both the thickness and the lateral size was conducted based on 56 sites of AFM images as shown in Figure S3. The result is shown in Figure 2b and discussed in the revised manuscript.

(4) In order to discuss the pathway of the prepared samples for gas permeability, the author should demonstrate the XRD of the nanosheets on AAO substrates.

Response: We thank the reviewer for this comment. As suggested by the reviewer, the XRD patterns of 2D MAMS-1 membranes supported on AAO substrates are provided in the revised Supplementary Information (Figure S10a) and main text (Figure 5a). The various crystal planes are also highlighted to visualize the gas pathways in MAMS-1 membranes (Figure S10b-d). The main gas pathway is PW1 with small aperture on (002) planes as described in the revised Supplementary Information: “The PXRD pattern of simulated MAMS-1 crystal features two peaks at 4.62 and 8.14 ° corresponding to (002), (100) and (011) crystal planes, respectively (Figure S10a). The (002) crystal plane is parallel to the basal plane of MAMS-1 nanosheets (Figure S10b), while the (100) and (011) crystal planes are almost perpendicular to it (Figure S10c and Figure S10d). In the case of the 40-nm membrane, only the peak from (002) crystal plane is detectable (Figure S10a), indicating the oriented stacking of MAMS-1 nanosheets along the basal plane exposing PW1 (Figure S10b) with smaller aperture suitable for molecular sieving gas separation.”

(5) Gas permeability results generally depend strongly on how the nanosheets sample prepared on the porous substrates and it is no possible to prepare the identical samples. The authors prepared three different types of sample (M1, M2 and M3). But these categorization is ambiguous. The authors should evaluate the error on the gas selectivity and transparency in each categorized sample.

Response: We agree with the reviewer that it is very challenging to obtain identical gas separation performance even under the same conditions. In order to give a clear categorization, we renamed the membrane samples as 4-nm membrane, 12-nm membrane, and 40-nm membrane. All the obtained separation performance of these membranes is listed in Table S2. As suggested by the reviewer, the averaged values of selectivity and permeance with errors are listed as well.

(6) *In order to understand the reversed thermal-switchable behavior seen in the nanosheets, the authors examined temperature dependence of PXRD on bulk crystals. It often seen that structural/phase change behavior is different between bulk materials and nanoscaled materials. The authors should demonstrate the PXRD measurements on exfoliated nanosheets.*

Response: We thank the reviewer for this comment. As suggested, we have demonstrated the temperature dependence of PXRD patterns for both the 40-nm membrane and the bulk MAMS-1 crystals as shown in Figure 5a and Figure 5b of the revised manuscript. The shift of (002) peak is identical between the 40-nm membrane and the bulk MAMS-1 crystals. The revision has been added to the main text as follows: “In order to confirm this, the 40-nm membrane and bulk MAMS-1 crystals were further characterized by in situ variable temperature PXRD (Figure 5a). Notably, the PXRD peak from the (002) crystal plane of the 40-nm membrane shifted toward higher two-theta angles from 4.64 to 4.92 ° upon heating from 20 to 100 °C. The peak remained almost unchanged upon further heating up to 200 °C, but started to shift back when the temperature was reduced below 80 °C and finally reached 4.64 ° at 30 °C. From 20 to 100 °C, a change of 0.28 ° toward higher two-theta angle was achieved in the (002) peak, indicating the contraction of lattice spacing between (002) planes from 1.9029 to 1.7946 nm based on the Bragg equation. A similar shift of the (002) peak upon heating/cooling was observed in the bulk MAMS-1 crystals (from 4.62 ° to 4.96 ° as shown in Figure 5b). Interestingly, the (100) and (011) peaks of bulk MAMS-1 crystals shift to 7.72 ° during heating, and return back to 8.14 ° upon cooling, corresponding to an expansion of 0.0589 nm in the ab crystalline planes and enlarged aperture size of PW1 at elevated temperatures (Figure 5c), which agrees well with the temperature-induced molecular-gating effects of MAMS-1 proposed by Zhou et al.³¹. However, although the aperture size of PW1 becomes larger at higher temperatures, its kinetic opening is still controlled by the rotation of tert-butyl groups. Preliminary MD simulation indicates that the MAMS-1 nanosheet is even impermeable to H₂ molecules if the free rotation of tert-butyl

groups is prohibited. The tert-butyl groups can rotate freely at room temperature because of the surrounding free volume. At higher temperatures, the free rotation of tert-butyl groups will be restricted due to the intensified steric hindrance caused by reduced interlayer distance, leading to blocked PW1 and sharply decreased gas permeance (Figure 5d)."

Reviewer: 2

This manuscript describes the fabrication of nanometer-thick 2D membranes displaying thermo-switchable molecular sieving CO₂/H₂ separation performance. The content is an interesting contribution where the two hot and timely concepts of 2D nanosheets and membrane-based gas separation are linked. However, there are issues here which would prevent the publication of this work in Nat. Comm..

Response: We thank the reviewer for this comment. We have thoroughly addressed the concerns raised by the reviewer. Major revisions have also been made in the revised manuscript according to the reviewer's suggestion. We hope the reviewer will be satisfied with these revisions.

(1) The authors reported the synthesis of bilayer-nanosheets from a MOF crystal and confirmed the structural features like porosity and thickness through several characterization methods. However, they did not make direct use of these bilayer-nanosheets for CO₂/H₂ separation because of the difficulties in the membrane fabrication process as they mentioned. Instead, they make the membrane with the thickness of 40 nm. Is the porosity of this 40 nm-thick membrane (M3-1) the same as that of the bilayer-4 nm-membrane? If not, what is the structure of these 40 nm-thick material on the AAO-substrate?

Response: We thank the reviewer for this comment. In the case of the 4-nm membrane and 12-nm membrane, the MAMS-1 nanosheets are relative flat as shown in Figure 3a and Figure 3b. However, some crumples can be clearly observed from the top-surface of the 40-nm membrane as shown in Figure 3c. As described in the main text, the capillary force should be the main driving force for the formation of 4-nm membrane layer: “In this study where porous AAO substrate was used, the solvent could evaporate and diffuse through the porous channels of the substrate. This process can exert a perpendicular capillary force to pull the fragile MAMS-1 nanosheets firmly toward the coarse AAO substrate causing rupture and pinholes of MAMS-1 nanosheets. Our speculation was further confirmed by the focused ion beam (FIB) TEM image of the cross-sectional area of the survived 4-nm membrane (Figure 3d), wherein a thin layer (ca. 4 nm) can be found concaving toward the porous channels of AAO substrate due to the capillary force²⁶. Given this condition, increasing the thickness of membrane layer should be able to preserve its integrity because the following MAMS-1 nanosheets covering on top of the first layer should experience less capillary force and thus may have a higher chance to survive.” However, the capillary force, which can allow a quick removal of solvent, would disappear once the AAO substrate is fully covered with MAMS-1 nanosheets. Therefore, the remaining solvent can only be removed by evaporation under such condition. Considering the high boiling point of the solvent used in this study (DMSO, b.p. = 189 °C), we speculate that the solvent molecules trapped in the central region of the 2D membranes cannot be removed as quickly as those located on the edge area, leading to crumples observed on the surface of the 40-nm membrane shown in Figure 3c. An illustration of this process is demonstrated in **Figure R2**. Thus, we believe the porosity of the 40-nm membrane layer should be larger than that of the 4-nm membrane. To better demonstrate the difference in membrane structure, an illustration of the 4-nm and 40-nm membrane structure is provided in **Figure R3**.

Figure R2. Illustration of the wrinkle formation process in the 40-nm membrane.

Figure R3. Membrane morphologies and illustrated structures of the 4-nm membrane (a-c) and 40-nm membrane (d-f).

(2) *Are the exfoliated bilayer nanosheets stacked again? Fig 3c and f may be not sufficient. The structural features of these 40-nm materials must be clarified since they are not bilayernanosheets (4 nm). The authors need to provide more information for these 40 nm-material besides XRD.*

Response: We thank the reviewer for this comment. The stacking of MAMS-1 nanosheets is a spontaneous process driven by the release of crystal lattice energy. This can be seen from

the growth of MAMS-1 crystals during hydrothermal reactions shown in **Figure R4a** (the SEM image is also shown as Figure 1a in the main text). We believe that the MAMS-1 nanosheets should re-stack together during membrane fabrication, which can be proven by the PXRD patterns. However, the restacked MAMS-1 nanosheets within the membrane layer are partially disordered as evidenced by the crumples on the membrane surface shown in Figure 3c. Such disordered stacking was also observed in other 2D nanosheets including ZIFs (*Science* **2014**, 346, 1356), graphene and graphene oxide (*Nat. Commun.* **2013**, 4, 2979). The disordered stacking of MAMS-1 nanosheets would lead to the formation of “amorphous-like” region within the membrane layer. A proposed structure of the 2D MAMS-1 membrane is presented in **Figure R4b**.

Figure R4. Structural features of MAMS-1 nanosheets within the bulk crystal (a) and the membrane layer (b).

(3) M3-1 for CO₂ separation (page 10, line 207 and line 222): The authors cannot use bulky MOF crystals to explain the separation on your nanosheets.

Response: We thank the reviewer for this comment. We agree with the reviewer that the performance of bulk MOF crystals can only give us a preliminary explanation of the membrane separation performance, which is summarized as follows in the revised manuscript: “A similar phenomenon was found in MOF-5 by molecular dynamics (MD) simulation, demonstrating a gas transport diffusivity sequence of H₂ > N₂ ≈ CH₄ > CO₂ due to the CO₂-philicity feature of this MOF⁴⁷. We expect a similar CO₂-philicity of MAMS-1 because of the highly polar internal surface of PW2 contributed by the hydrophilic octanickel [Ni₈(μ₃-OH)₄] clusters³¹, which is confirmed by the gas sorption isotherms (Figure S12) and different adsorption heat between CO₂ (23.0 – 31.2 kJ mol⁻¹) and H₂ (3.3 – 6.5 kJ mol⁻¹, Figure S13). Therefore, CO₂ molecules with a higher quadrupole moment (4.3 × 10²⁶ esu cm²) should be trapped more strongly within PW2 than the other gases, leading to diffusion-controlled permeation along PW2 which is unfavorable for CO₂⁴⁸.” We understand that the gas separation in membranes is not only determined by the gas-membrane affinity (adsorption equilibrium), but also the gas diffusion speed within membranes (diffusion kinetics). Therefore, the molecular dynamics (MD) simulation was further conducted to obtain more details on the diffusion behavior of gas molecules through 2D MAMS-1 membranes. The simulation system composed of bilayered MAMS-1 nanosheets is shown in **Figure R5a**. A snapshot of the molecule distribution after 80 ns is shown in **Figure R5b**. The following description is included in the revised main text: “A similar phenomenon was also observed for the simulation using an equimolar H₂/CO₂ mixture. 45 % H₂ molecules permeated through the bilayered nanosheet, while the CO₂ molecules penetrated merely into PW2 of the first layer (Figure 4c and Figure S19, a video based on the simulation of H₂/CO₂ separation is provided in Supplementary Information). The simulation results strongly support our previous conclusion that low CO₂ permeance is contributed by the

molecular sieving effects of the narrow PW1 aperture as well as the retarded diffusivity through PW2 in MAMS-1 nanosheets.”

Figure R5. Simulation system for the permeation of an equimolar H_2/CO_2 mixture through a bilayered MAMS-1 nanosheet (a) and snapshot after 80 ns of simulation (b, also shown as Figure 4c in the main text). An equimolar mixture of H_2/CO_2 (40 molecules in total) and a vacuum are on the left and right of the nanosheet, respectively. A graphene plate is exerted to separate the feed and permeate chambers. Colour of the atoms: C, cyan; Ni, blue; O, red; H, white.

(4) Please measure the CO_2/H_2 gas uptake performance for the target-membrane (M3-1), like what was reported in Ref. 21, to get a better understanding of these significant separation differences.

Response: We thank the reviewer for this comment. We agree with the reviewer that gas uptake of membranes is very useful to understand their separation performance. However, the small amount of 2D MAMS-1 membrane layer prevents us from reaching a convincing conclusion by directly measuring the membranes. Generally speaking, the weight of the AAO substrate in a membrane is approximate 5 mg, which is much higher than that of the

MAMS-1 membrane layer. As shown in **Figure R6a**, the BET surface area of AAO substrate alone is $83 \text{ m}^2 \text{ g}^{-1}$, which is very close to that of the exfoliated MAMS-1 nanosheets ($126.1 \text{ m}^2 \text{ g}^{-1}$) as mentioned in Figure 2c in main text. Therefore, the gas uptake of the membrane should be significantly contributed by the AAO substrate because of the large weight difference between the substrate and membrane layer. Alternatively, the gas uptake of the exfoliated MAMS-1 nanosheets should be much more reasonable to understand the membrane separation performance as reported in Ref 21. As shown in **Figure R6b**, the gas uptakes for CO_2 and H_2 at 298 K 1 atm in MAMS-1 nanosheets are $26.7 \text{ cm}^3 \cdot \text{g}^{-1}$ and $18.0 \text{ cm}^3 \cdot \text{g}^{-1}$, respectively. The isotherms indicate the preferential adsorption of CO_2 over H_2 in both bulk MAMS-1 crystals and exfoliated MAMS-1 nanosheets, which is helpful to understand the retarded diffusion of CO_2 over H_2 in PW2 of 2D MAMS-1 membranes.

Figure R6. (a) Nitrogen sorption isotherm at 77 K for AAO substrate. (b) CO_2 and H_2 uptakes of bulk MAMS-1 crystals (open symbols) and exfoliated MAMS-1 nanosheets (closed symbols) at 25 °C.

(5) *In addition, are all exfoliated nanosheets two layers (4 nm)? Or this is average number?*

Response: We thank the reviewer for this comment. A statistical analysis of both the thickness and the lateral size was conducted based on 56 sites of AFM images as shown in

Figure S3. The result is shown in Figure 2b and described in the revised manuscript as follows: “As demonstrated by the atomic force microscopy (AFM) analyses on a total of 56 sites, more than 95 % of them have a thickness of *ca.* 4 nm, proving the efficient exfoliation of MAMS-1 crystals into bilayered MAMS-1 nanosheets. However, the lateral size distribution of exfoliated MAMS-1 nanosheets is rather broad (4.7 - 24.3 μm , with an averaged value of $10.7 \pm 4.8 \mu\text{m}$, Figure 2b and Figure S3), which may not be suitable for membrane fabrication and further purification is needed (*vide infra*).”

(6) *How about the XRD structure of the 2-layers nanosheets? The N_2 -adsorption curves look unmoral in comparison with that of ZIF-sheets (Ref. 13) and CuBDC-MOF-sheets (Ref. 21). Why? What is pore size distribution of these Nanosheets? This is important to understand the gas adsorption and diffusion.*

Response: We thank the reviewer for this comment. As suggested, we have prepared 2-layered nanosheets (4-nm membrane), 6-layered nanosheets (12-nm membrane), and 20-layered nanosheets (40-nm membrane) on AAO substrates for XRD characterization. The results are shown in Figure S10a and the discussion has been added into the revised Supplementary Information as follows: “In the case of the 40-nm membrane, only the peak from (002) crystal plane is detectable (Figure S10a), indicating the oriented stacking of MAMS-1 nanosheets along the basal plane exposing PW1 (Figure S10b) with smaller aperture suitable for molecular sieving gas separation. On the contrary, no PXRD peak can be detected from the 4-nm membrane and the 12-nm membrane (Figure S10a). Considering the identical fabrication procedure between the membranes, the missing PXRD peak of the 4-nm membrane and the 12-nm membrane can be attributed to the ultra-small thickness of the membranes which prevents the effective X-ray scattering for detectable PXRD signals.” We also have observed the difference in N_2 -adsorption curve compared to ZIF-sheets and CuBDC-MOF-sheets. Such difference may be attributed to the board size distribution of MAMS-1 nanosheets. As shown in Figure 2b and Figure S3, the lateral size of MAMS-1

nanosheets ranges from 4.7 μm to 24.3 μm . We have also added this observation in the revised manuscript as follows: “As demonstrated by the atomic force microscopy (AFM) analyses on a total of 56 sites, more than 95 % of them have a thickness of *ca.* 4 nm, proving the efficient exfoliation of MAMS-1 crystals into bilayered MAMS-1 nanosheets. However, the lateral size distribution of exfoliated MAMS-1 nanosheets is rather broad (4.7 - 24.3 μm , with an averaged value of 10.7 ± 4.8 μm , Figure 2b and Figure S3), which may not be suitable for membrane fabrication and further purification is needed (*vide infra*).” Meanwhile, the pore size distributions of exfoliated MAMS-1 nanosheets and bulk MAMS-1 crystals are calculated using DFT model based on N_2 adsorption data. As shown in **Figure R7**, the smallest pore sizes of bulk MAMS-1 crystals and exfoliated MAMS-1 nanosheets are 17.6 nm and 31.5 nm, respectively, which should be contributed by the interstitial voids among particles. No micropores inside MAMS-1 crystals or nanosheets can be identified, possibly due to the small pore size and 2D structure that prevent the penetration of probe N_2 molecules. This result agrees well with the originally reported N_2 sorption isotherms of MAMS-1 (*Angew. Chem., Int. Ed.* **2007**, *46*, 2458).

Figure R7. Pore size distribution of bulk MAMS-1 crystals (a) and exfoliated MAMS-1 nanosheets (b) based on the DFT calculation.

(7) For solvent-selective sedimentation approach, the authors suggest that the high dipole moment of DMSO is a key factor. More experiments with different solvents are needed to confirm this suggestion.

Response: We thank the reviewer for this comment. Unfortunately we do not have strong evidence to support the conclusion yet. Although we believe the dipole moment might be important, another important factor is solvent density. **Table R1** lists the density of solvents used in this study, which ranges from 0.655 g cm⁻³ to 1.10 g cm⁻³. Since the calculated density of MAMS-1 is 1.482 g cm⁻³, solvents with appropriate density may help to stabilize larger MAMS-1 nanosheets. We have removed the statement regarding the effect of dipole moment in the revised manuscript to avoid misleading.

Table R1. Parameters of the organic solvents used in this study.

Solvent	Density / g cm ⁻³	Dipole moment×10 ¹⁸ / esu·cm
Hexane	0.655	0
Methanol	0.792	1.70
Acetonitrile	0.786	3.45
DMF	0.944	3.86
DMSO	1.10	3.90

(8) Page 8 line 164, what is the thickness of M2 membranes? Reference section: Pay attention to the abbreviation (several typos)

Response: We thank the reviewer for this comment. As suggested, the thickness of M2 membranes (renamed as 12-nm membrane in the revised manuscript) was determined by FIB-TEM to be 12 nm (Figure 3e). We have carefully checked the reference section to remove the typos.

Reviewer 3

This work reported an ultrathin bilayered MAMS-1 membrane for the gas separation of H₂/CO₂ pair. The exfoliated MAMS-1 nanosheets were purified by a novel selective sedimentation approach, which may give the hints for other researchers worked on 2D membranes. The thermo-responsive property of parent MOF materials gives the obtained 2D membrane a thermal-switchable behavior on the gas separation. This gating effect of membrane materials may open the window of multi-function membrane separation. This is a state-of-the-art work. It will be a significant contribution to this field and will be of great interest to a broad readership of Nature Communications. I have several minor comments before its publication.

Response: We really appreciate the positive comment from this reviewer. All the questions and concerns raised by the reviewer have been clarified as follows.

(1) As the author mentioned, larger pieces of MAMS-1 layer were obtained in this work, so how about the mechanical strength?

Response: We thank the reviewer for this comment. Actually, we have been trying to measure the mechanical strength of exfoliated MAMS-1 nanosheets by AFM indention experiments. Prior to the measurement, the MAMS-1 nanosheets were drop-casted onto a 10 mm × 10 mm chip which is identical to the membrane fabrication procedure. The chip was designed with patterns of different dimensions of 0.5 μm and 1 μm (**Figure R8**). To our disappointment, the bilayered MAMS-1 nanosheets were fractured by the capillary force during the solvent evaporation, indicating a poor mechanical strength of the bilayered MAMS-1 nanosheets. Based on the calculation, the capillary forces originated from the

evaporation of hexane, ethanol, DMSO, and water in 1.0 μm pores at room temperature are as high as 56.3 nN, 70.1 nN, 137.5 nN, and 228.6 nN, respectively. The force needed to rupture a 8-layered MOF nanosheet was measured to be approximate 40 nN (*Chem. Sci.* **2015**, 6, 2553), which confirms the difficulty in measuring the mechanical strength of bilayered MAMS-1 nanosheets.

Figure R8. SEM image of the wafer substrate used for fracture force test.

(2) *The purification of the 2D nanosheets is interesting. How about the diameter distribution of the nanosheets, and a larger scale AFM image contained several pieces of MAMS-1 layers should be better. And if the relationship between the purification yield and time could be studied and supplied, it will be very help to understand the separation process of this method.*

Response: We thank the reviewer for this comment. In order to visualize the lateral size distribution of the exfoliated MAMS-1 nanosheets, we conducted the statistical analysis on 56 sites of AFM images (Figure 2b) with the description in the main text as follows: “As demonstrated by the atomic force microscopy (AFM) analyses on a total of 56 sites, more than 95 % of them have a thickness of *ca.* 4 nm, proving the efficient exfoliation of MAMS-1 crystals into bilayered MAMS-1 nanosheets. However, the lateral size distribution of exfoliated MAMS-1 nanosheets is rather broad (4.7 - 24.3 μm , with an averaged value of $10.7 \pm 4.8 \mu\text{m}$, Figure 2b and Figure S3), which may not be suitable for membrane fabrication and further purification is needed (*vide infra*).” The exfoliation rate or yield was measured by weighing AAO substrates before and after drop-casting. The yield calculated from the MAMS-1 suspension after centrifugation is $13.05 \pm 1.60 \%$. The yield became 6.5 % after solvent sedimentation. This result has been added in the revised Supplementary Information as follows: “In the case of hexane, the exfoliation rate was about 6.5 % with a concentration of 0.065 mg mL^{-1} for the suspension measured by weighing AAO substrates before and after drop-casting.”

(3) *The single gas test part should be moved to the ahead of the first paragraph of mixture gas test part to let the manuscript flow. In the scheme picture of gas test setup (Figure S9), there will be a gas path of Ar connected to GC, if the TCD was used as detector. It will be better to sum up more information of all the membranes in Table S1, such as membrane thickness and gas test condition.*

Response: We thank the reviewer for this comment. All the membrane performance was firstly evaluated by the separation of equimolar H_2/CO_2 mixture at room temperature. Thereafter, we optimized the fabrication condition for membrane preparation. In order to better understand the separation mechanism of 2D MAMS-1 membrane, the single gas permeation was further measured using the 40-nm membrane. A gas path of carry gas (Ar)

for GC is shown in the scheme picture of gas test setup (Figure S11). The membrane thickness is also added in Table S2.

(4) There is a mistake in Table S2. The performance of JUC-150 listed in the table is not right. (It is the data of another membrane with a longer pillar of 4,4-bipyridine one).

Response: We thank the reviewer for this comment. The separation performance of JUC-150 membrane has been revised according to the literature. In addition, all the separation performance listed in Table S3 (previously labeled as Table S2) has been double checked.

Reviewers' Comments:

Reviewer #1 (Remarks to the Author)

The authors revised the manuscript according to the reviewers comments including new results of additional experiments. The topic is hot and of interests for diverse readers. The reviewer now recommends the manuscript to be published in Nature Communications.

Reviewer #2 (Remarks to the Author)

This is an interesting paper reporting the use of exfoliated 2D materials for gas separation. The rebuttal letter is quite comprehensive and addresses well some of the concerns of reviewer 1. Although the method for producing exfoliated 2D materials via thermal expansion is not new, the separation properties of the resulting membranes for H₂/CO₂ are outstanding. It can be published after consideration of the following two issues:

(1) What are the separation properties for the mixture gas of CO₂ and N₂ using these membranes made of the 2D materials? The preferential uptake of CO₂ should block the transport of N₂ and thereby leads to high selectivity for CO₂. I hope that the authors can conduct such an experiment to further demonstrate the unique performance of these membranes.

(2) The recent paper using the gas expansion for exfoliation of 2D materials (BN) should be cited (Zhu, W. S.; et. al. Controlled Gas Exfoliation of Boron Nitride into Few-Layered Nanosheets. *Angew. Chem.-Int. Edit.* 2016, 55, 10766-10770).

Reviewer 1

The authors revised the manuscript according to the reviewers comments including new results of additional experiments. The topic is hot and of interests for diverse readers. The reviewer now recommends the manuscript to be published in Nature Communications.

Response: We thank the reviewer for this positive comment.

Reviewer: 2

This is an interesting paper reporting the use of exfoliated 2D materials for gas separation. The rebuttal letter is quite comprehensive and addresses well some of the concerns of reviewer 1. Although the method for producing exfoliated 2D materials via thermal

expansion is not new, the separation properties of the resulting membranes for H₂/CO₂ are outstanding. It can be published after consideration of the following two issues:

(1) What are the separation properties for the mixture gas of CO₂ and N₂ using these membranes made of the 2D materials? The preferential uptake of CO₂ should block the transport of N₂ and thereby leads to high selectivity for CO₂. I hope that the authors can conduct such an experiment to further demonstrate the unique performance of these membranes.

Response: We thank the reviewer for this insightful suggestion. However, according to the single gas permeation results (Figure 4a in the main text), the 2D MAMS-1 membranes in this study are much more permeable to H₂ or He over CO₂ or N₂. Even though the permselectivity for N₂/CO₂ is 2.18, the N₂ permeance is merely 6.76 GPU, which is impractical for real gas separations. We speculate that the N₂ permeance would be further reduced in the mixture gas of CO₂ and N₂ because of the preferential adsorption of CO₂. Therefore, the technical and economic feasibility would be the major problem for N₂/CO₂ separation using 2D MAMS-1 membranes in this study.

(2) The recent paper using the gas expansion for exfoliation of 2D materials (BN) should be cited (Zhu, W. S.; et. al. Controlled Gas Exfoliation of Boron Nitride into Few-Layered Nanosheets. Angew. Chem.-Int. Edit. 2016, 55, 10766-10770).

Response: We thank the reviewer for this valuable suggestion. The recommended paper can further elaborate the mechanism of the freeze-thaw exfoliation approach in this study. Therefore, we have cited the recommended paper and added the following discussion in the revised manuscript: “Recently, Zhu *et al.*³⁸ reported thermal-expansion-triggered gas exfoliation of bulk *h*-boron nitride based on their expansion and curling triggered by the huge temperature variation.”

Reviewer 3

****Reviewer #3 submitted comments to the editor only. They stated that all their comments had been addressed and they recommended publication.****

Response: We thank the reviewer for the recommendation.

Thank you very much for your consideration of this manuscript.